# DIVIDE AND EXPLORE: MULTI-AGENT SEPARATE EXPLORATION WITH SHARED INTRINSIC MOTIVATIONS

## ABSTRACT

One of the greatest challenges of reinforcement learning is efficient exploration, especially when training signals are sparse or deceptive. The main difficulty of exploration lies in the size and complexity of the state space, which makes simple approaches such as exhaustive search infeasible. Our work is based on two important observations. On one hand, modern computing platforms are extremely scalable in terms of number of computing nodes and cores, which can complete asynchronous and well load-balanced computational tasks very fast. On the other hand, Divide-and-Conquer is a commonly used technique in computer science to solve similar problems (such as SAT) of doing efficient search in extremely large state space. In this paper, we apply the idea of divide-and-conquer in the context of intelligent exploration. The resulting exploration scheme can be combined with various specific intrinsic rewards designed for the given task. In our exploration scheme, the learning algorithm can automatically divide the state space into regions, and each agent is assigned to explore one of these regions. All the agents run asynchronously and they can be deployed onto modern distributed computing platforms. Our experiments show that the proposed method is highly efficient and is able to achieve state-of-the-art results in many RL tasks such as MiniGrid and Vizdoom.

## 1 INTRODUCTION

Deep reinforcement learning is a very important learning framework to train intelligent agents to solve complex tasks (Sutton & Barto, 2018). Despite recent success stories in a variety of real tasks (Silver et al., 2016; Vinyals et al., 2019; Lee et al., 2020), the difficulty of exploration remains a major challenge for DRL when the reward is sparse or deceptive (Burda et al., 2018). Recent research in exploration has made significant progresses thanks to the idea of intrinsic motivation (Schmidhuber, 2010). Intrinsic motivation could be viewed as objectives for unsupervised reinforcement learning (Levine, 2020), and their goal is usually not to guide the agent to complete a specific task, but to help the agent to discover interesting states and trajectories which could help the agent to understand the environment and find possible target solutions for specific tasks. Many previous works on intrinsic motivation (Raileanu & Rocktäschel, 2020; Burda et al., 2018; 2019) can be understood as proposing some kind of metrics to measure the novelty of the states, and then using the metric as an intrinsic reward to encourage agents to reach novel states, such as count-based exploration (Bellemare et al., 2016) and ICM (Pathak et al., 2017).

Despite of the significant progresses that have been made, in theory, efficient exploration is still very challenging if not impossible, due to the infeasible size of the state space to be explored. Similar difficulties are usually seen in the state-space search problems in theoretical computer science. For example, in SMT or SAT problems (De Moura & Bjørner, 2008), one is required to search for a feasible solution to satisfy certain constraints, and these problems are usually NP complete because of the exponential size of the state space.

A very common class of practical algorithms to solve these search problems is divide and conquer. The algorithm first tries to divide the state space into certain pieces and then search for results in each piece of state space. A significant benefit of this approach is it can be easily modified to take advantage of modern distributed/concurrent computing hardware, because it can naturally divide a hard problem into smaller and simpler ones, allowing different computing nodes to complete

the tasks asynchronously. Divide-and-conquer type of algorithms are among the state-of-the art algorithms to solve SMT or SAT problems.

In this paper, we take inspiration from divide and conquer algorithms in theoretical computer science and try to adapt them to the exploration problems in RL. Different from traditional exploration mechanisms with a single exploring agent, we train a number of concurrent exploring agents. We carefully design our intrinsic reward such that each exploring agent is encouraged to explore only one region in the state space. In other words, the state space is automatically divided into several components, where each agent is responsible for exploring one of these components.

In order to achieve this, we first choose a reward function based on intrinsic motivation, e.g. count-based reward or ICM. This function is able to output a novelty score for a given transition. We initialize $K$ agents, each agent is equipped with its own version of reward function. Then we design a multi-party intrinsic reward for each agent consisting of two parts: (1) a reward to discourage the agent to visit states which has already been explored by other agents. (2) a self exploration reward to encourage the agent to visit novel states which has not been visited by itself. Agents run in a fully asynchronous manner, no agent has to wait for other other agent's progress to take exploration steps. All agents communicate with each other through shared memory, under the abstraction of a simple read-write shared object, which can be made lock-free, i.e., the learning system can always progress regardless of some computing nodes may delay or fail. These properties make our algorithm highly scalable. Our experiments show that the proposed method is highly efficient and is able to achieve state-of-the-art results in many RL tasks such as MiniGrid and Vizdoom.

## 2 RELATED WORK

**Intrinsic Motivation** The study of intrinsic motivation has been of interest for some time (Schmidhuber, 2010). Earlier works on intrinsic motivation tended to analyse with psychology and combined with reinforcement learning in tabular setting (Deci et al., 1981; Singh et al., 2005). With the breakthroughs of deep reinforcement learning, there has been a surge of interest in applying intrinsic motivation with deep neural networks for hard exploration RL problems, in which count-based exploration and curiosity-driven exploration are two main promising research branches. For count-based methods, the agent is encouraged to explore rarely visited states, while for curiosity-driven methods, the agent is encouraged to explore via learning the world model. **Count** (Bellemare et al., 2016) extends count-based methods to non-tabular setting by deriving pseudo-counts with a density model. Ostrovski et al. (2017) adapts Bellemare et al. (2016) with PixelCNN. **RND** (Burda et al., 2018) uses random network distillation to measure the familiarity of states, and **RGU** (Badia et al., 2020) combines **RND** with an episodic novelty module. **ICM**(Pathak et al., 2017) builds a forward dynamic model and an inverse dynamic model to measure the curiosity for exploration. **RIDE** (Raileanu & Rocktäschel, 2020) has similar networks with ICM while using a different intrinsic reward combined with episodic state visitation counts. **AMIGo** (Campero et al., 2020) proposes a meta-learning framework for generating adversarially motivated intrinsic goals in curriculum to guide agent exploration. **RAPID** (Zha et al., 2021) ranks past trajectories with episodic scores and does imitation learning for distillation. **AGAC** (Flet-Berliac et al., 2021) formulates an actor-critic framework with adversary. Our work is different from above by building a multi-agent learning mechanism to accelerate exploration.

**Distributed Reinforcement Learning** The scalability of large scale distributed learning in DRL is a main challenge for reinforcement learning. Most of previous works scale up deep reinforcement learning with distributed asynchronous SGD (Dean et al., 2012) by distributing RL components. Gorila (Nair et al., 2015) scales DQN (Mnih et al., 2015) with distributed Q networks and replay buffers. A3C (Mnih et al., 2016) uses multiple workers for training and collects all gradients for parameter synchronous. Unlike A3C, IMPALA (Espeholt et al., 2018) transfers all trajectories to multiple central learners for distributed learning. Ape-X (Horgan et al., 2018) consists of a distributed replay buffers and a synchronous learner for value-based methods. SEED RL (Espeholt et al., 2019) and OpenAI Rapid (OpenAI, 2018) apply central GPU based inference for acceleration. For most of the above works, the main parallelism exploited by the distributed framework is data parallelism, e.g. they use multiple nodes for collecting data, and divide minibatches among computing nodes and aggregate on the gradients computed by each node. Our model, on the other hand, can be viewed as taking advantage a new parallelism specific to RL problems — state space parallelism.

Instead of dividing minibatches for the nodes, we let the agents divide the state space into pieces, which greatly improves exploration efficiency.

# 3 BACKGROUND

## 3.1 PROBLEM FORMULATION

In this paper, we consider an RL problem formulated as a Markov Decision Process(MDP), which is defined as a tuple MDP $= \{S, A, P, R, \gamma\}$, where $S$ is the state space, $A$ is the action space of the agent, $P$ is the transition probabilities which represents the stochastic transition function $S \times A \rightarrow S$, $R(s, a)$ is the reward function after the agent executes an action, $\gamma$ is the discount factor. The goal is to maximize the expected accumulated rewards $R = E(\sum_{t=0}^{T} \gamma^t r(s_t, a_t))$ the agent receives when executing series of optimal actions in episodes with policy $\pi$.

In our method, other than the rewards $r_t$ received from environment at timestep $t$, our agents also receive a intrinsic reward $R_t$ as an guidance for efficient exploration. The intrinsic reward is a parameterized function of state and action $R_t(s, a) = f_\theta(s, a)$, where the parameters $\theta = \theta(M)$ depends on the agent's history or replay buffer $M$. For example, in count-based reward, $\theta$ is the number of visits at each state.

Instead of a single exploring agent, our proposed method trains $K$ independent exploring agents. The exploring agents do not communicate or affect each other directly, however they indirectly affect each other through the intrinsic rewards. Each agent uses its own history or replay memory to train its own version of $f_\theta$(the intrinsic motivation model). For any agent, the total intrinsic reward used for agent training is formed by combining all trained reward functions of all the agents.

## 3.2 CONCURRENT SHARED OBJECTS

A commonly used primitive in concurrent or distributed computing is concurrent shared objects. It is basically the user interface of concurrent data structures. A concurrent shared objects is composed of several methods which all processes can call. The behavior of a shared object is defined by its sequential execution traces. For example, a concurrent Set object has three methods, add($v$), find($v$) and remove($v$). The behavior of this concurrent object is defined by the sequential executions of an ordinary (non-concurrent) Set data type. Also, a property called "linearizability" is applied to guarantee consistent usage of the concurrent object. Linearizability basically says that for any concurrent execution traces of the concurrent object, one can find a sequential execution trace which yields exactly the same results and the sequential execution trace preserves the real time ordering of the concurrent execution trace. In other words, all the methods of the concurrent objects can be viewed as executed atomically. In this paper, we assume all concurrent objects are linearizable.

# 4 DIVIDE AND EXPLORE

We design a learning mechanism named Divide-and-Explore (D&E) to realize the idea mentioned above. An overview of D&E is illustrated in Fig.1. The learning framework consists of $n$ independent agents $\{A^1, A^2, \ldots, A^n\}$, and each agent tries to explore a particular region in the state space. We let replay memory $M^i$ denote the set of all the past experiences of the $i$th agent $A^i$ in the form $(s_t^i, a_t^i, r_t^i, s_{t+1}^i)$. Let $M = (M^1, \cdots M^n)$ be the collection of replay memories of all agents. Note that in practice we usually don't need to store all the past experiences in the replay memory, but some sufficient statistics of the experiences is enough.

Our algorithm first specifies a learnable intrinsic motivation model $f_\theta$ as a reward function, and $\theta$ is the model's parameters. For each agent $A^i$, we keep track of a distinct model $f^i = f_{\theta^i}$, where $\theta^i = \theta(M^i)$ is a function of the replay memory of agent $A^i$. When agent $A^i$ takes a transition $(s_t^i, a_t^i, r_t^i, s_{t+1}^i)$, for every agent $A^j$, our training architecture computes a intrinsic reward $\hat{r}_t^{i,j}$ based on the transition and the agent's $f^j$, as following

$$\hat{r}_t^{i,j} = f^j(s_t^i, a_t^i, r_t^i, s_{t+1}^i) \tag{1}$$

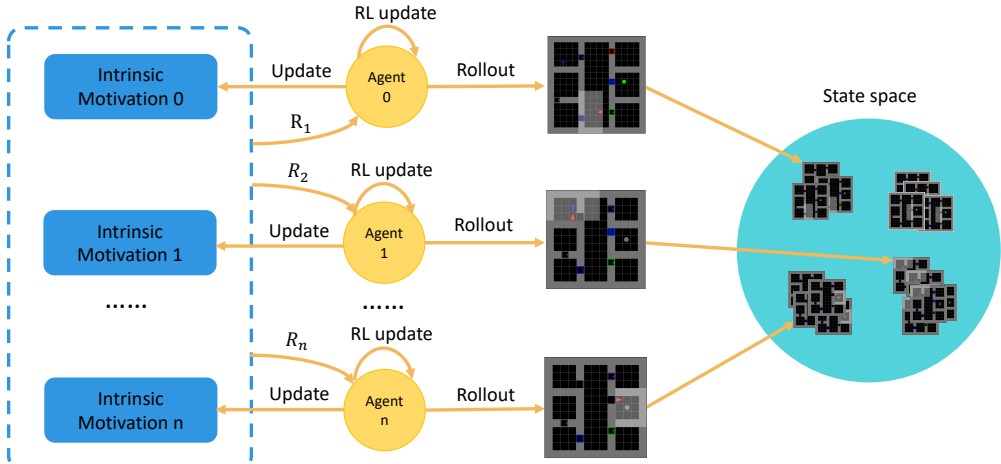

Figure 1: An Overview of Divide and Explore. Agents are exploring different regions of state space automatically, and intrinsic motivations are shared across all the agents for updating.

Intuitively, $\hat{r}_t^{i,j}$ measures the transition's novelty for agent $A^j$. If $\hat{r}_t^{i,j}$ is large, it means that the state $s_{i+1}$ is rarely visited and not explored fully by Agent $A^j$. The intrinsic motivation model $f_\theta$ can be selected with any kind of count-based, curiosity-driven and memory-based exploration methods (e.g. Pathak et al. (2017); Burda et al. (2018)) according to the specified task.

## 4.1 DIVIDE AND EXPLORE REWARD

Here we define the D&E intrinsic reward for agent learning as below:

$$R_t^i = \omega_o \sum_{j \neq i} \hat{r}_t^{i,j} + \omega_s \hat{r}_t^{i,i} \tag{2}$$

where $\omega_o(n-1) + \omega_s = 1$.

According to the definition, the D&E reward aggregates other agents' intrinsic rewards and the agent's own intrinsic reward, which encourages the agent to explore the state space where all agents have rarely explored. This mechanism enables full use of all agents' experience and gets rid of wasting time on fully explored state space so as to improve exploration efficiency. The weights $\omega_o$ and $\omega_s$ determine the exploration preference between agent separability and self exploration. If $\omega_o$ is zero, the algorithm turns out to be conventional intrinsic-driven exploration, while if $\omega_s$ is zero, the algorithm only considers separability and the agent may converge to a local state space where other agents haven't visited and stop exploring further. In practice, we set $\omega_o > \omega_s$ to encourage agents to explore separately while keeping self exploration.

With the D&E intrinsic reward, the total reward is defined as follows:

$$\tilde{r}_t^i = r_t^i + \alpha_t^i R_t^i \tag{3}$$

where $r_t^i$ is the task reward provided by the environment, and $\alpha_t^i$ is the weight of D&E intrinsic reward. Base on the definition above, we are able to turn a sparse reward task into a dense one, and apply conventional model free reinforcement learning algorithms such as PPO (Schulman et al., 2017) or IMPALA (Espeholt et al., 2018) to train the agent policy.

## 4.2 AUTOMATIC REWARD WEIGHT DECAY FOR EXPLOITATION

While the main goal of Divide-and-Explore is to encourage efficient exploration, we also consider the trade-off between exploration and exploitation. In conventional model free reinforcement learning, the trade-off is usually controlled in the agent roll out process via $\epsilon$-greedy (Mnih et al., 2015)

or the action probability distribution (Mnih et al., 2016). Because in D&E we use intrinsic reward to encourage agent to explore, we reduce the intrinsic reward weight to make agent pay more attention to the trajectories with high extrinsic reward, especially when the agent has trained a lot and has a probability of obtaining some rewards. The decay of intrinsic reward weight can be formulated as $\alpha = d_t \cdot \alpha_0$, and $d_t$ is decay factor. Compared to linear decay, we design an adaptive decay method as follows:

$$\alpha_t^i = d_t^i \alpha_0 = \max\{1 - \frac{1}{U}(\sum_{j=0}^{N} \phi^{N-j} \mathcal{R}_j^i), 0\} \cdot \alpha_0 \tag{4}$$

in which $\alpha_0 > 0$ is the initial value of $\alpha_t^i$, $d_t^i$ is the decay factor of agent $A^i$ at time $t$, $N$ is the number of $A^i$'s historical episodes and $\mathcal{R}_j^i$ is the return of episodes. We consider more about newly generated episodes by introducing $\phi \in (0, 1]$. $U$ is a hyperparameter, which determines the decay rate, and in general, we let $U = \sum_{j=0}^{N} \phi^{N-j} \mathcal{R}^*$. According to Eq.4, $\alpha_t^i$ has negative correlation with the sum of historical rewards and is adjusted automatically during training, which matches the goal of trade-off.

### 4.3 FULLY ASYNCHRONOUS UPDATES WITH CONCURRENT DATA STRUCTURES

In order to speed up computing and scale our approach to large number of exploring agents, we use non-blocking concurrent computing primitives in our implementation. In our algorithm, the reward functions $f^i = f_{\theta^i}, i = 1, 2 \cdots n$ is shared across all agents. Only agent $A^i$ is allowed to modify $\theta^i$, but all agents can read $\theta^i$. We can store $\theta = (\theta^1, \theta^2, \cdots \theta^n)$ in a shared concurrent object $O$.

In the terminology of concurrent data structures, we define $O$ as a concurrent object that allows two operations, $\text{read}(i)$ which returns a copy of $\theta^i$ and $\text{update}(i, v)$ which overwrite current $\theta^i$ into value $v$. A lot of concurrent data structures could be used to implement this concurrent object, such as read-write locks (locking), transactional memory or software transactional memory (lock-free). In our work, we use software transactional memory since it is non-blocking, making our algorithm fully asynchronous.

Our approach is summarized in Algorithm 1. All the agents are trained asynchronously in a distributed training framework, and the shared object $O$ is read every $L$ steps. Besides, there is a special attribute of D&E that all agents will be convergence as long as only one agent accomplishes the task. With the shared intrinsic motivation and learning mechanism, the failure experience of other agents is transferred to the succeeding agent as bad cases to avoid repetition of wrong exploration, which demonstrates the superiority of this algorithm.

## 5 EXPERIMENTS

In this section, we evaluate D&E on two challenging reinforcement learning research platforms **MiniGrid** (Chevalier-Boisvert et al., 2018) and **VizDoom** (Kempka et al., 2016), and both of them have hard exploration environments with sparse rewards, which are commonly used as benchmarks. For all our experiments, we compare our approach against various state-of-the-art baselines for hard exploration including **Count** (Bellemare et al., 2016), **ICM** (Pathak et al., 2017), **RND** (Burda et al., 2018), **AMIGo** (Campero et al., 2020), **RIDE** (Raileanu & Rocktäschel, 2020), **RAPID** (Zha et al., 2021) and **AGAC** (Flet-Berliac et al., 2021).

### 5.1 MINIGRID EXPERIMENTS

**Environment**    The MiniGrid environments are a set of procedurally-generated 2D grid worlds with sparse reward. Following **AGAC** (Flet-Berliac et al., 2021)'s setting, we choose three types of tasks for experiments: MultiRoom(**MR**), KeyCorridor(**KC**) and ObstructedMaze(**OM**). In the MultiRoom tasks, there are multiple rooms connected one-by-one with doors. The agent has to go through these rooms to reach a goal. In the KeyCorridor tasks, several rooms are nested together and the mission of the agent is to find a key so as to open a locked room and pick up an object. For the last, the ObstructedMaze tasks are updated versions of KeyCorridor and most challenging where

---

**Algorithm 1** Divide and Explore

---

1: **Input:** Time horizon $T$, decay frequency $K$, reward model update frequency $L$, reward weight $\omega$ and $\alpha_0$
2: Initialize $n$ agent policies $\{\pi^1, \pi^2, ..., \pi^n\}$ and the shared object contains reward model parameters $O$, agent replay memory $\{M^1, M^2, ..., M^n\}$. For each agent policy, initialize a local copy of reward models $\hat{\theta}^1, \hat{\theta}^2, \cdots \hat{\theta}^n$.
3: **for** iteration=1,2,... until convergence **do**
4:     **for all** agents $A^i, i = 1, 2 \cdots n$ **do** asynchronously
5:         Roll out $T$ timesteps with $\pi^i$ in the environment
6:         **for** every $L$ steps **do**
7:             Update local reward model parameters $\hat{\theta}^j = O.\text{read}(j), j \neq i$
8:         **end for**
9:         Compute total rewards using Eq.3 based on locally stored $\hat{\theta}^1, \hat{\theta}^2, \cdots \hat{\theta}^n$.
10:         Update $\pi^i$ with RL algorithm like PPO
11:         Update agent replay memory $M^i$
12:         Compute new model parameters based on the replay memory $\hat{\theta}^i = \theta^i(M^i)$
13:         Update shared object by calling shared method $O.\text{update}(i, \hat{\theta}^i)$
14:     **end for**
15:     **for** every K steps **do**
16:         Update reward weight $\alpha$ with Eq.4
17:     **end for**
18: **end for**
19: Return the optimal policy $\pi^*$

---

the key is hidden in boxes and the door is obstructed by balls. For all tasks, we make use of partial observations, which means the observation space is a 7x7x3 tensor indicating the information of partially-observable grid cells, and there are 7 actions for executing.

**D&E with Count-based Exploration**    Due to the simplicity of 2D grid environment, we count the states in the form of tabular as exploration bonus in the MiniGrid tasks, which has been used in previous works, e.g. **Count** (Bellemare et al., 2016), **RIDE** (Raileanu & Rocktäschel, 2020) and **AGAC** (Flet-Berliac et al., 2021). Therefore, We combine count-based exploration with D&E for experiments. In detail, we define the intrinsic reward by:

$$\hat{r}_t^{i,j} = 1 / \sqrt{\sum_{s \in M^j} I(s = s_{t+1}^i) + 1} \tag{5}$$

where $I(\cdot)$ is indicator function. We also introduce an inner-episodic count mask as a multiplier by replacing original reward weight $\alpha_t^i$ with:

$$\alpha_t^i \leftarrow I(\sum_{s \in C_i} I(s = s_t^i) = 0)\alpha_t^i \tag{6}$$

where $C_i$ is the trajectory in current episode. With this mask, if a state has been visited in the trajectory, the intrinsic reward will be masked, which improve the exploration efficiency.

**Training**    We use PPO (Schulman et al. (2017)) as a standard model free RL algorithm to train all the agents. For fair comparison, the architecture of actor and critic network we use is the same as **AGAC**'s network in Flet-Berliac et al. (2021), which consists of 3 convolutional blocks and a fully connected layer for feature processing. More concretely, each convolutional block contains a 32-filter 3x3 convolution with stride size 2, and the hidden size of fully connected layer is 512. The number of agents in D&E is set to 3 as default for all tasks. More hyperparameters can be seen in Appendix A.

**Results**    We train D&E with various MiniGrid tasks and compare the results with baselines in both exploration efficiency and performance. For exploration efficiency, we illustrated the learning

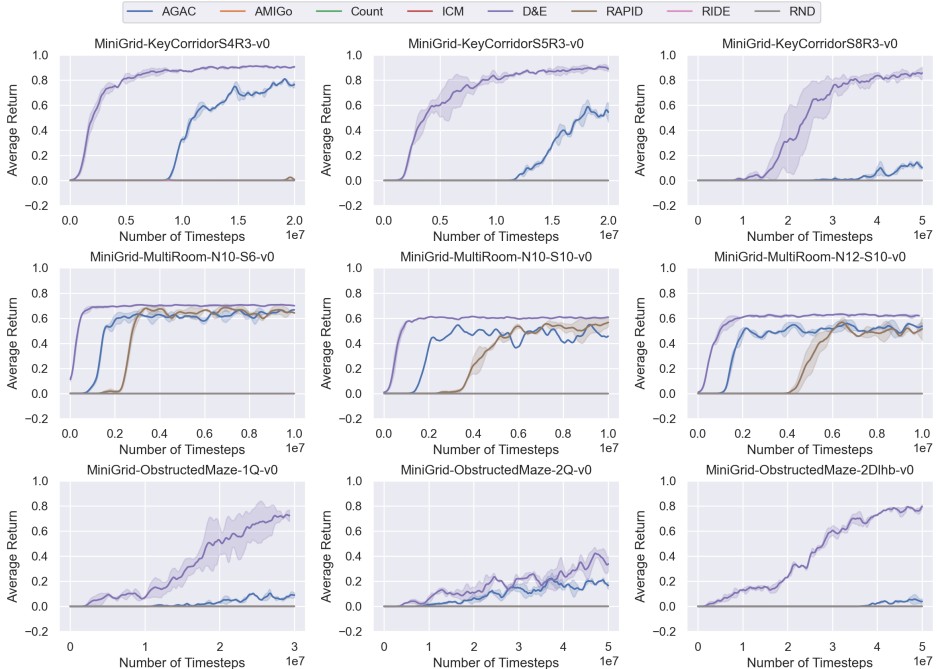

Figure 2: Performance of D&E and baselines in MiniGrid. We plot the best agent's performance in D&E.

| Task | KC-S4R3 | KC-S5R3 | MR-N10S10 | MR-N12S10 | OM-2Dlhb | OM-1Q | OM-2Q |
|---|---|---|---|---|---|---|---|
| D&E | 0.91 | **0.90** | **0.60** | **0.60** | **0.16** | **0.39** | 0.09 |
| AGAC | **0.93** | **0.90** | 0.52 | 0.51 | 0.04 | 0.09 | **0.20** |
| RAPID | 0.90 | 0.00 | 0.54 | 0.55 | 0.00 | 0.00 | 0.00 |
| RIDE | 0.00 | 0.00 | 0.00 | 0.00 | 0.00 | 0.00 | 0.00 |
| AMIGo | 0.00 | 0.00 | 0.00 | 0.00 | 0.00 | 0.00 | 0.00 |
| RND | 0.00 | 0.00 | 0.00 | 0.00 | 0.03 | 0.00 | 0.00 |
| ICM | 0.00 | 0.00 | 0.00 | 0.00 | 0.00 | 0.00 | 0.00 |
| Count | 0.00 | 0.00 | 0.00 | 0.00 | 0.00 | 0.00 | 0.00 |

Table 1: Final performance of D&E and baselines on MiniGrid environments. All the methods are trained in 50M total timesteps.

curves of D&E and baselines in less than 50M timesteps in Fig.2. Because of the training concurrency in all D&E's agents, we record the best agent's performance for D&E with the agent's timesteps by default. In all the tasks, D&E achieves state-of-the-art results and the performance boosts with much less timesteps compared to other methods especially AGAC and RAPID. Only in the ObstructedMaze-2Q task, the performance of AGAC is comparable with D&E.

For performance only, we report the final performance of all methods in Table 1. All baselines are trained in 50M timesteps. Due to the multi-agent learning mechanism, the total number of timesteps used in D&E is directly proportional to the number of agents. Therefore, we only record the performance of D&E at 16.7M timesteps for a fair comparison. As we can see in the table, D&E performs best in most scenarios while several approaches such as RND, ICM and Count totally failed in all the listed tasks.

For better understanding the exploration efficiency, we also plot the state visitation heatmaps of the challenging MultiRoom-N12-S10 task with 1M timesteps in Fig.3. The policies are trained in a singleton environment. While D&E finds a good trajectory to the last room with few wrong actions, AGAC spends more time in the journey, and RAPID is stuck in the sixth room.

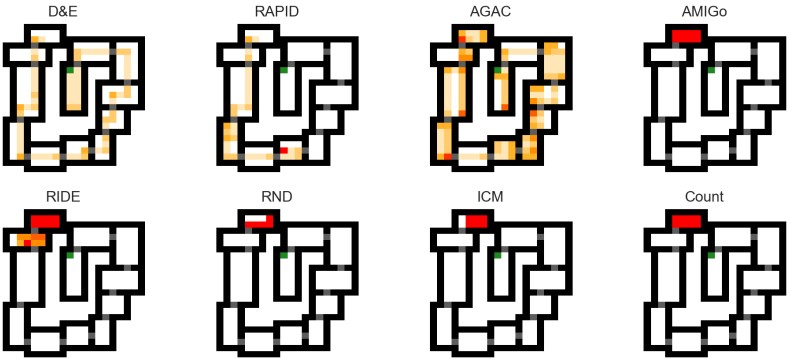

Figure 3: State visitation heatmaps in the MultiRoom-N12-S10 task. All the comparison methods only train 1M timesteps.

| Timesteps | 1M | 2M | 3M | 4M | 5M | 6M |
|---|---|---|---|---|---|---|
| D&E | **0.23** ± 0.5 | **0.96**± 0.1 | **0.97**± 0.02 | **0.97**± 0.01 | **0.97**± 0.001 | **0.97**± 0.001 |
| AGAC | 0.12± 0.5 | 0.84± 0.1 | 0.95± 0.1 | **0.97**± 0.01 | **0.97**± 0.001 | **0.97**± 0.001 |
| RIDE | ≤ 0. | ≤ 0. | ≤ 0. | ≤ 0. | ≤ 0. | 0.95± 0.001 |
| ICM | ≤ 0. | ≤ 0. | ≤ 0. | ≤ 0. | ≤ 0. | 0.95± 0.001 |
| Rapid | ≤ 0. | ≤ 0. | ≤ 0. | ≤ 0. | ≤ 0. | ≤ 0. |
| AMIGo | ≤ 0. | ≤ 0. | ≤ 0. | ≤ 0. | ≤ 0. | ≤ 0. |
| RND | ≤ 0. | ≤ 0. | ≤ 0. | ≤ 0. | ≤ 0. | ≤ 0. |
| Count | ≤ 0. | ≤ 0. | ≤ 0. | ≤ 0. | ≤ 0. | ≤ 0. |

Table 2: Performance of D&E and baselines on the VizDoom environment.

## 5.2 VizDoom Experiments

**Environment**    The VizDoom environments are a series of Doom-based 3D scenarios for reinforcement learning. We choose the MyWayHome task following **AGAC** (Flet-Berliac et al., 2021)'s setting. The MyWayHome task consists of multiple rooms and corridors, which is a 3D navigation task that the agent explores rooms to find the object. Different from MiniGrid, the state is an ego-centric visual observation with a size of 42x42x4, and the number of action is 5.

**D&E with Curiosity-Driven Exploration**    In order to demonstrate the generality of D&E and compare with baselines, we get rid of using count-based reward and combine D&E with the curiosity-driven exploration method **ICM** (Pathak et al., 2017). Every agent has its independent ICM which calculates intrinsic reward based on transition $(s_t, a_t, s_{t+1})$. For each agent, the ICM is trained together with its policy and shared across all the agents. We set the learnable reward model $f^i$ mentioned in Eq.1 as the ICM model, following **ICM**'s definition with normalization.

**Training and Results**    Following MiniGrid experiments, we train D&E in VizDoom with minor changes. We use the same network architecture of actor and critic with different state input in VizDoom, but replace count-based exploration with **ICM** using the original implementation. The hyperparameters are reported in Appendix A. We list the comparison results in Table 2 and D&E outperforms other baselines in any selected timesteps. Interestingly, while ICM is slow to solve the task, D&E learns fast and achieves high performance. Even when we compare D&E at 2M timesteps (6M timesteps in total agents) with ICM at 6M timesteps, the performance of D&E is still better.

## 5.3 Ablation Study

In the ablation study, we aim to answer following questions: 1) Is D&E capable to guide agents to explore different regions of state space separately? 2) Compared to single-agent setting, is D&E's

learning mechanism a key reason to improve exploration efficiency? 3) How the number of agents influence the performance?

For question 1, we train D&E on the KeyCorridorS6R3 task with singleton environment, and the number of agents is 3. The exploration process of different agents is shown in Fig.4, in which state visitation count in historical trajectories is accumulated and visualized in the form of heatmap. As we can see in the figure, it is clear that three agents are exploring different regions of the environment separately, which validates the learning mechanism of D&E.

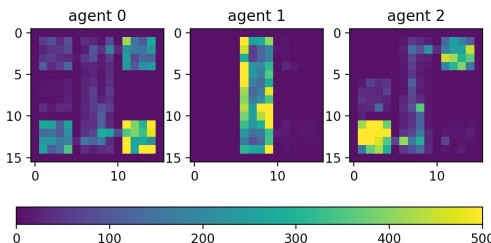

Figure 4: Exploration processes of three different agents in singleton environment.

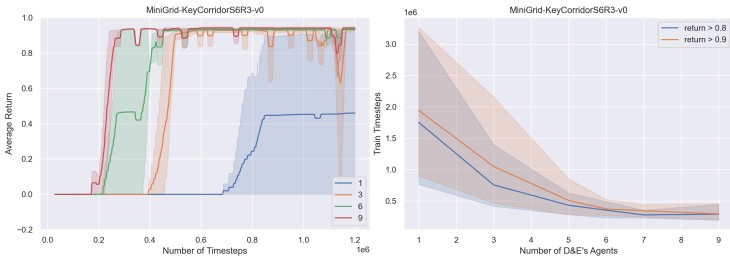

Figure 5: The performance of D&E with different number of agents.

For question 2 and 3, we train D&E on the KeyCorridorS6R3 task with various number of agents in singleton environment. The left figure in Fig.5 shows the best agent's learning curve under different agent number setting, which confirms that the number of agent brings benefits to the training efficiency of the algorithm. In fact, if the number of agents is one, the method is a conventional count-based method. On the other hand, as shown in the right figure in Fig.5, we also plot the number of training timesteps required when the return of episode reaches 0.8 and 0.9 for the first time. While the more agents, the less time algorithm needs to be trained, it shows a diminishing marginal effect. Thus, the number of agents should be chosen according to the task.

## 6   CONCLUSIONS

We proposed a brand new algorithm named Divide-and-Explore(D&E) to handle the hard exploration problem in reinforcement learning. Inspired by divide-and-conquer, D&E trains multiple concurrent exploring agents, and successfully guides each agent exploring different regions of state space with shared intrinsic motivations while keeping exploring the boundary, which produces state-of-the-art results. More generally, D&E can be seen as a framework, which can combine existing single-agent exploration methods, e.g. Count, ICM, RIDE and so on, to improve learning efficiency. In the future work, we'd like to extend this idea to large scale setting and test it on general RL benchmarks like atari games (Mnih et al., 2015) and MuJoCo tasks (Todorov et al., 2012). And we will also explore the application of D&E on 3D navigation in robotics (Zhelo et al., 2018) or playtesting coverage in games (Gordillo et al., 2021).

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

## A    HYPERPARAMETERS

The hyperparameters of D&E used in both MiniGrid and VizDoom environments are listed in Table 3. For Count, RND, ICM, RIDE, we conduct experiments based on the authors' implementation[1], while the policy networks are modified to be the same as D&E. And for other algorithms, we use the authors' implementations with default hyperparameters, e.g. AMIGo[2], RAPID[3] and AGAC[4].

| Parameter | Value |
|---|:---:|
| Time horizon $T$ | 2048 |
| Number of epochs | 4 |
| Number of minibatches | 8 |
| Learning rate | 3e-4 |
| Final learning rate | 3e-5 |
| Learning rate anneal schedule | linear |
| Learning rate anneal steps | 1e4 (3e4 in VizDoom) |
| Discount $\gamma$ | 0.99 |
| GAE parameter $\lambda$ | 0.95 |
| Gradient clip threshold | 1.0 (5.0 in VizDoom) |
| Number of agents | 3 |
| Decay frequency $K$ | 1 |
| Reward model update frequency $L$ | 2048 |
| Reward weight $\omega_o$ | 0.465 |
| Reward weight $\omega_s$ | 0.07 |
| Reward weight $\alpha_0$ | 0.1 |
| Threshold $U$ | 42 (30 in VizDoom) |
| Decay rate $\phi$ | 0.98 |

Table 3: Hyperparameters used in D&E

## B    ADDITIONAL EXPERIMENTS

### B.1    HARDER EXPLORATION TASK

There are many harder exploration tasks in MiniGrid that have not been well resolved in previous studies, for example, MiniGrid-KeyCorridorS10R4-v0. The difficulty of this task comes from three perspectives. Firstly, because the agent can only observe a small range of $7 \times 7$, large map will confuse corridors and rooms, making it difficult for the agent to know about its own location through observation. Secondly, the larger environment makes agent's trajectory longer and the rewards become more sparse, and this reinforces agent's dependence on intrinsic rewards during training. Thirdly, the larger environment makes the connection between rooms more complex, and agent often needs to go through several rooms to find the key and the object. However, D&E achieves amazing performance as shown in Fig.6, which shows that D&E does enhance ability to solve problems rather than simply improve efficiency.

### B.2    EXPERIMENT DETAILS

To the results mentioned in Fig.2, we plot the best performance in several independent runs of baseline algorithms, to illustrate the benefits are mainly from the particular intrinsic motivation. As shown in Fig.7, compared with independent runs, the interact between different agents and intrinsic reward calculated from all agents' replay buffer indeed accelerate the exploration efficiency.

---

[1]https://github.com/facebookresearch/impact-driven-exploration

[2]https://github.com/facebookresearch/adversarially-motivated-intrinsic-goals

[3]https://github.com/daochenzha/rapid

[4]https://github.com/yfletberliac/adversarially-guided-actor-critic

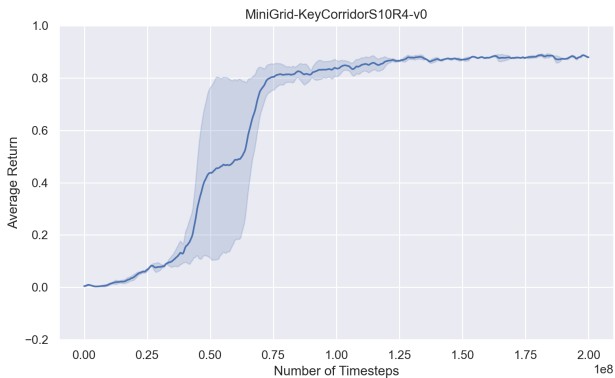

Figure 6: The performance of D&E with different decay method.

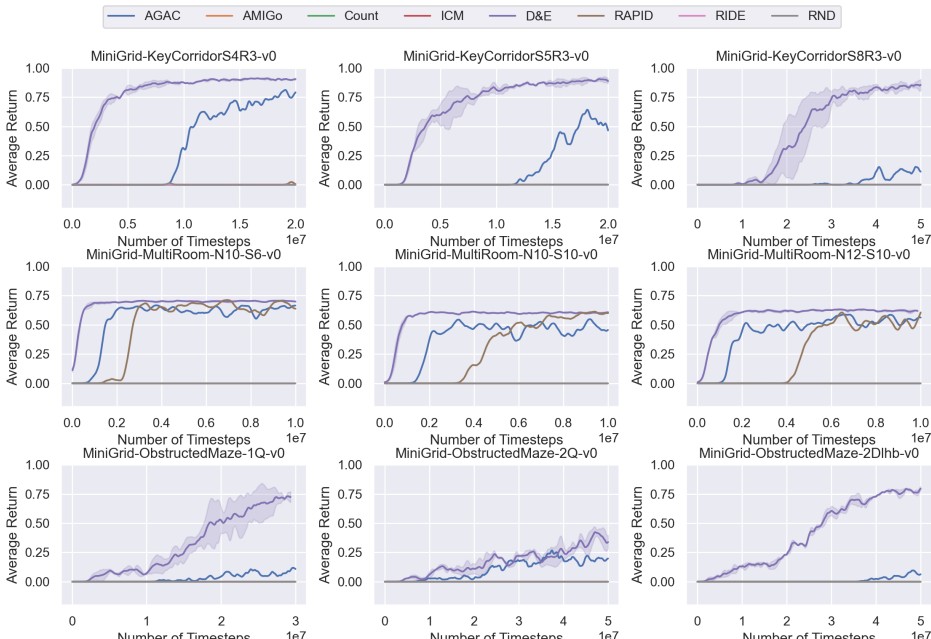

Figure 7: Performance of D&E and baselines in MiniGrid. We plot the best agent's performance in D&E and best performance in several independent runs of baseline algorithms.

### B.3 ABLATION EXPERIMENTS FOR DECAY METHOD

As a supplement, we also compare the effects of different intrinsic reward weight decay methods. We compare the conventional linear decay and adaptive decay in the MiniGrid-KeyCorridorS6R3-v0 task. The intrinsic reward weight in the former is calculated as:

$$\alpha_t^i = \max\{1 - \frac{N}{U}, 0\} \cdot \alpha_0 \tag{7}$$

while in the latter it is calculated as Eq.4. The results are shown in the left figure of Fig.8, which validates the benefits of the adaptive decay method. Considering the historical episodes' rewards, agent can make trade-off between exploration and exploitation better, and reduce exploration when it has found good trajectories and obtained high extrinsic rewards, which is similar with human decision. We plot the curve of decay factor $d_t$, and there are three phases in the training process.

At the beginning, agent rarely find a trajectory to complete the task and the decay factor $d_t$ is large which means that agent should pay more attention to intrinsic reward and explore as much as possible. When train timesteps is larger than 2M, agent has an increasingly high probability of completing tasks, and $d_t$ decreases fast. After 10M training, agent's policy converges and it does not need to explore.

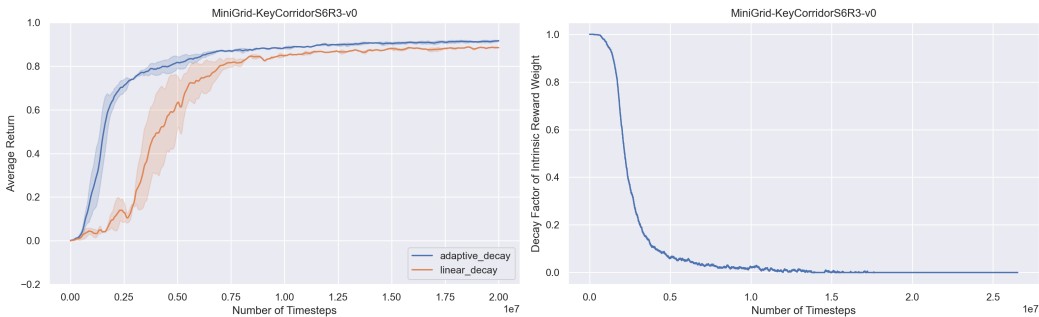

Figure 8: The performance of D&E with different decay methods.

