# OpenReview forum: "Divide and Explore: Multi-Agent Separate Exploration with Shared Intrinsic Motivations"
_ICLR.cc/2022/Conference — ICLR 2022 Submitted_

### Official Review · Reviewer_VtVT · 2021-11-01

**Correctness:** 4
**Technical Novelty And Significance:** 2
**Empirical Novelty And Significance:** 3
**Recommendation:** 5
**Confidence:** 5

**Main Review:**

## Strengths
* The setting of concurrent multi-agent exploration for single agent tasks is worthwhile and understudied
* Thorough evaluation with respect to existing single agent exploration methods

## Weaknesses
* Seemingly missing references/discussion regarding some extremely relevant related work. Most closely related is [1] as it addresses the same setting of multiple concurrent agents solving a single-agent task by a divide-and-conquer exploration approach. Moreover, [2] introduces the idea of multiple agents each maintaining their own "novelty detection" function and combining these functions to generate intrinsic rewards that prevent redundant exploration of similar spaces.
* It's unclear that the D&E reward will always result in the desired behavior. It seems as though it requires agents to redundantly explore the same regions or else $\sum_{j \neq i} \tilde{r}^{i,j}$ will always be non-zero.
* The inner-episodic count mask seems like it would not have a scalable analogue in more complex domains.
* The method referred to as "Count" in the experiments is a psuedo-count method which derives an approximate visit count from a learned density model. A worthwhile baseline (for only the MiniGrid setting) would be to train with the same "real" count-based intrinsic rewards without concurrent exploration and the D&E reward. This would eliminate the confounding factor of the effectiveness of the individual intrinsic reward mechanism and highlight how important the contributions of this paper are.

## Questions
* How are timesteps being counted for your method? I assume it's the sum of steps across all concurrent processes?
* Why not plot the training curve for VizDoom rather than include a table?


### References
[1] Dimakopoulou, Maria, and Benjamin Van Roy. "Coordinated exploration in concurrent reinforcement learning." International Conference on Machine Learning. PMLR, 2018.

[2] Iqbal, Shariq, and Fei Sha. "Coordinated Exploration via Intrinsic Rewards for Multi-Agent Reinforcement Learning." arXiv preprint arXiv:1905.12127 (2019).

**Summary Of The Paper:**

This paper presents an approach to enable parallelizable "divide-and-conquer" style exploration in sparse reward RL tasks, where each concurrent process maintains its own intrinsic reward function but they are combined in a way that minimizes redundant exploration of the state space across processes.

**Summary Of The Review:**

This paper presents interesting ideas; however, they are extremely similar to prior work that is not discussed or compared to in the paper. It is difficult to judge the significance of this work without comparisons to (or at the very least discussion of) this prior work.

---

> ### Author Response · Authors · 2021-11-21
> **Response to Reviewer VtVT**
>
> We sincerely thank reviewer VtVT for the feedback and suggestions on our paper. We address the reviewer's concerns below.
>
> Q: “Seemingly missing references/discussion regarding some extremely relevant related work.”
>
> A: We read the two papers you mentioned. The first article is focused on concurrent reinforcement learning, where all the agents are assumed to all information between each other. While in our setting, we train each agent independently and only intrinsic motivations are shared for intrinsic reward calculation. Even though we also use Divide-and-Conquer as inspiration, the particular idea and implementation of D&E is different with seed sampling.
>
>  The second article is to solve exploration problems in the multi-agent environment, while we focus on the single-agent environment by flexibly expanding the number of agents to improve the efficiency of the algorithm, and the number of agents is variable according to our needs.
>
>
> Q: “It's unclear that the D&E reward will always result in the desired behavior.”
>
> A: Ideally, due to $\omega_o>\omega_s$, we tend to avoid agent exploring states already explored by other agents. However, with the randomness of the environment and agents' actions, we hardly find that agents get stuck in local regions in practice.
>
> Q: “The inner-episodic count mask seems like it would not have a scalable analog in more complex domains.”
>
> A: The inner-episodic count mask is only used in MiniGrid environments, whose state space is discrete. In complex domains, we can use other intrinsic reward models, for example, ICM on Vizdoom. We used the mask to combine episode state count and inner-episodic count, while other intrinsic reward models may be trained by history data.
>
> Q: “A worthwhile baseline (for only the MiniGrid setting) would be to train with the same "real" count-based intrinsic rewards without concurrent exploration and the D&E reward.”
>
> A: We are sorry that we didn’t describe it clearly. We conducted experiments based on the implementation in https://github.com/facebookresearch/impact-driven-exploration, where Count is a simple count-based reward, with agent’s observation as key. In addition, although we didn’t conduct experiments for all environments when agent num is 1, it is equivalent to the experiment without considering D&E reward.
>
> Q: “How are timesteps being counted for your method? I assume it's the sum of steps across all concurrent processes?”
>
> A: We used the sum of steps across all concurrent processes in Table 1 and the steps of the best agent in other results.
>
> Q: “Why not plot the training curve for VizDoom rather than include a table?”
>
> A: We wanted to show the results quantitatively, and thus wrote the specific rewards obtained by the best agent at different stages of the training process. This setting is the same with AGAC.

---

### Official Review · Reviewer_xEFb · 2021-11-02

**Correctness:** 3
**Technical Novelty And Significance:** 3
**Empirical Novelty And Significance:** 3
**Recommendation:** 5
**Confidence:** 4

**Main Review:**

Strengths

- This paper is clearly written. The central ideas around the novel multi-agent intrinsic reward and concurrent implementation details are clearly communicated. The authors also highlight where they introduce additional hyperparameters.
- The experimental setting consists of standard exploration benchmarks—MiniGrid and VizDoom.
- A comprehensive set of intrinsic reward baselines used in procedurally-generated environments is included in the experiments.
- The method is simple and the results are strong in comparison to the intrinsic reward baselines.

Weaknesses

- The D&E method is only benchmarked for the case when the intrinsic reward is a simple count-based reward. Therefore, the experiments contain no direct ablation of the key multi-agent intrinsic reward that is the crux of D&E. This is because the Count method introduced by Bellemere et al, 2016 is not the same as simply applying a count-based exploration bonus. Including an ablation using only the simple count-based intrinsic reward would make the results more conclusive.
- Related to the above point, including results for D&E combined with at least one of the non-count-based intrinsic motivation methods would further strengthen the results in support of D&E's multi-agent intrinsic reward.
- Including information about how the additional hyperparameters introduced by D&E were tuned and how sensitive the method is to these parameters would be useful. This is especially important as D&E introduces several hyperparameters: number of agents, decay frequency $K$, reward model update frequency $L$, reward weights $w_0$, $w_s$, $\alpha_0$, threshold $U$, and decay rate $\phi$.
- The authors mention that they used the default hyperparameters for the baseline methods, but as some of the methods were not originally benchmarked on VizDoom (e.g. AMIGo and RAPID), it seems that these VizDoom baselines may be improperly tuned.
- Including additional seeds for Figure 4 and averaging over seeds for Figure 3 would add confidence that these results are not cherrypicked.

**Summary Of The Paper:**

This paper introduces a method for performing multi-agent exploration by supplementing any intrinsic reward method by providing each agent an additional reward that is the sum of intrinsic rewards of all other agents were they to experience the agent's transition. This additional weighted reward term allows the agent to balance exploring states that other agents have rarely encountered alongside those that the agent iself has rarely encountered. The experiments show that this divide-and-conquer strategy for multi-agent exploration results in state-of-the-art results on MiniGrid and VizDoom.

**Summary Of The Review:**

This paper clearly presents a simple multi-agent intrinsic reward method for encouraging agents to not only explore regions that they have rarely encountered, but also those that the other agents have rarely encountered. Their results are promising, but lack proper baselines to get the full story about the effectiveness of their method. If the authors address the weaknesses pointed out above, I will enthusiastically raise my score to an 8.

---

> ### Author Response · Authors · 2021-11-21
> **Response to Reviewer xEFb**
>
> We sincerely thank reviewer xEFb for the feedback and suggestions on our paper. We address the reviewer's concerns below.
>
> Q: “Including an ablation using only the simple count-based intrinsic reward would make the results more conclusive.”
>
> A: We are sorry that we didn’t describe it clearly. We conducted experiments based on the implementation in https://github.com/facebookresearch/impact-driven-exploration, where Count is a simple count-based reward, with agent’s observation as key.
>
> Q: “Including information about how the additional hyperparameters introduced by D&E were tuned and how sensitive the method is to these parameters would be useful.”
>
> A: About the hyperparameters, we list some here.
>
> 1. Agent num. We tested a different numbers of agents and observed the effect of the number on the experimental effect. As shown in Figure5, we finally selected 3 as the default parameter.
>
> 2. Intuitively, it is clearly optimal to update the decay weights in real-time, so the default is to update the weights every step, i.e. K=1, although this imposes some computational costs.
> 3. For L, we tested several values and found little impact. We simply used the default value of 2048 in AGAC.
> 4. For the weights $\alpha$, we used 0.1, which is commonly used in previous studies, as the weight of IR. For $\omega_s$ and $\omega_o$，we experimented with different ratios $\omega_o/\omega_s$, and found that $\omega_o/\omega_s=10$ worked better. In fact, the ratio $\omega_o/\omega_s$ means that how many agents should take into account the exploration histories of others. When it is infinite, agents focus on states that have not been explored by other agents, and conversely, other agents’ explorations are not considered.
> 5. When the agent's policy network converges, we can assume that $R^i_j$ in Equation 4 is a constant $R$, then the decay item in Equation 4 can be rewritten as $\frac{R}{(1-\phi)U}$. In MiniGrid, we chose 0.84 as a threshold, which means that when extrinsic rewards agent obtains are greater than 0.84, the agent does not need to explore. Then when $\phi=0.98$, $U=42$, we have $\frac{R}{(1-\phi)U}=1$. The parameter $\phi$ determines how much importance we place on past rewards.

---

> > ### Comment · Reviewer_xEFb · 2021-11-24
> > **Thanks for the response**
> >
> > Thanks for the clarifications. The RIDE count baselines use episodic count-based exploration rewards. Does the D&E method used in the experiments also keep to episodic counts or use batch or lifetime counts? The latter variations have been shown to solve most MiniGrid exploration environments alone, so it is important to make sure you are not using those for your D&E implementation.
> >
> > Thanks also for the additional comments on hyperparameters. It would be useful to add this discussion to the appendix. I believe testing in a second environment, e.g. VizDoom, would instill further confidence in these intuitions and understandings of how the hyperparameter settings impact performance.
> >
> > The authors have only responded to two of the five pieces of feedback I provided. The other points remain unacknowledged, and so presently, I will keep to my original score.

---

### Official Review · Reviewer_BfC2 · 2021-11-02

**Correctness:** 2
**Technical Novelty And Significance:** 1
**Empirical Novelty And Significance:** 2
**Recommendation:** 3
**Confidence:** 4

**Main Review:**

Strengths:
* The idea of parallelizing agents for exploration is interesting.
* The approach can solve many tasks on MiniGrid, which are challenging due to their procedural nature and sparse rewards.
* The approach can solve the VizDoom MyWayHome task, which many baselines (Rapid, AMIGo, RND, Count) cannot.

Weaknesses:
* The method is very similar to prior work. See “Efficient Exploration via State Marginal Matching” (Lee et al. 2019) section 2.3, “Better SMM with Mixtures of Policies.” D&E is like SMM Equation 3 terms (b) and (d). D&E can also be seen as “Diversity Is All You Need” (Eysenbach et al. 2018), where each D&E agent is a DIAYN skill. Additionally, if we set $w_o=w_s=1/n$ in Equation 2 (weighting current agent and other agents equally), D&E is equivalent to a general intrinsic reward. I’d expect at least an ablation with different weights of $w_o$ and $w_s$.
* The results are not apples-to-apples comparisons. Figures 2 and 5 and Tables 1 and 2 should show the total number of steps across all agents for each method. Since D&E uses 3 agents, its performance is effectively reported as being 3 times more sample efficient than it actually is.
* Using 3 agents in parallel is not very scaled. For example, IMPALA is frequently run with 40+ actors. If the motivation for D&E is parallelism, then I’d expect more agents to be used.

Other thoughts and suggestions:
* Section 4.2: Versions of learning reward weights (as described in Equation 4) have been done before. See “On Multi-objective Policy Optimization as a Tool for Reinforcement Learning” (Abdolmaleki et al. 2021) for example. I also don’t understand the described reasoning of matching the “goal of tradeoff” – what does this concretely refer to?
* The count mask defined in Equation 6 should be ablated and/or used in baselines.
* It’s unclear whether the D&E heatmap in Figure 3 shows just the best of the 3 agents or the average. I’m guessing it’s just the best agent since the normalization looks different (the D&E heatmap is lighter than the others). If this is the case, it would be nice to include heatmaps for the other agents as well.
* Section 5.2: It’s odd to use a count-based reward on MiniGrid and switch ICM on VizDoom. Pick one to use consistently across these settings or show both formulations everywhere.
* Section 5.3: The experimental questions aren’t that interesting. As a rule of thumb, yes/no questions aren’t great for these.
* Figure 4: These heatmaps are interesting and shed a lot more light on the method. I’d be curious to see these heatmaps for MultiRoom (the environment in Figure 3) because it’s less clear to me what each agent would cover in this more sequential environment.

**Summary Of The Paper:**

The paper proposes looking at intrinsic exploration from a multi-agent perspective. The proposed method (D&E) rewards an agent for visiting unseen states and for visiting states that are unseen by other agents. The motivation is that if there are multiple parts of the state space, agents can explore these in parallel, thus speeding up the wall-clock exploration time. Experiments look at D&E performance on several MiniGrid tasks as well as VizDoom.

**Summary Of The Review:**

This is an interesting setting, but the method lacks novelty (see first weakness). Additionally, the results do not paint an accurate picture of D&E compared with other baselines (see second weakness). Perhaps this project could look more at reconciling options/skill learning work with intrinsic motivation and do more evaluation and comparison of baselines in these areas.

---

> ### Author Response · Authors · 2021-11-21
> **Response to Reviewer BfC2**
>
> We sincerely thank reviewer BfC2 for the feedback and suggestions on our paper. We address the reviewer's concerns below.
>
> Q: "The method is very similar to prior work."
>
> A: The main novelties of D&E are: 1. We designed a method to transform a single-agent environment into a multi-agent environment, and each agent has its own intrinsic reward model. 2. Taking advantage of parallel computing to accelerate the exploration process. The idea of splitting state space is similar, but D&E constructs intrinsic reward by aggregating each agent’s intrinsic reward model, which makes it possible to apply some method from previous studies, such as ICM, and it can be seen as using an ensemble approach to get the novelty of state. Compared with SMM, D&E does not need a target state distribution as input.
>
> Q: "If we set $\omega_o=\omega_s=1/n$ in Equation 2, D&E is equivalent to a general intrinsic reward."
>
> A: We agree with you. When the weights are the same, different agents just speed up the data production process. Considering that if different agents reach the same state, they will get the same reward, and their policy networks will also converge.
>
> Q: "Figures 2 and 5 and Tables 1 and 2 should show the total number of steps across all agents for each method.”
>
> A: We also considered how to better present the experimental comparison results, and there are two reasons. 1. It is unreasonable to directly use all timesteps of D&E when plotting the best agent’s performance, for its training process does not directly use data from other agents. 2. We also want to highlight the advantage of D&E in terms of time, i.e., the time benefits from parallel training. So we still used the true timesteps of the best agent in D&E when plotting the curves, while doing the emphasis in Table 1.
>
> Q: "Using 3 agents in parallel is not very scaled."
>
> A: We believe that parallelism is an important aspect to improve efficiency. However, in D&E, each agent contains both worker and learner, and using a large number of agents for a simple environment will lead to a waste of computational resources. We checked the effect of the different numbers of agents on the efficiency improvement and found that while the more agents, the less time algorithm needs to be trained, it shows a diminishing marginal effect. Besides, the training process of each agent is independent and can be combined with algorithms such as IMPALA.
>
> Q: "I also don’t understand the described reasoning of matching the 'goal of trade-off' – what does this concretely refer to?"
>
> A: Our "goal of trade-off" refers to the trade-off between exploration and exploitation during the training process. By reading the article you mentioned, we found that perhaps our description led to confusion. In the article you mentioned, the trade-off refers to the weighting between multiple objectives, which is the importance of different objectives, while we expect agents to gradually reduce exploration and increase exploitation during training.
>
> Q: “It’s odd to use a count-based reward on MiniGrid and switch ICM on VizDoom.”
>
> A: We want to create a general framework where D&E is compatible with the intrinsic reward model from previous studies, such as ICM and RND. Considering that the simple count method isn’t suitable for continuous state spaces, and ICM performs well on vizdoom environments, so we switched ICM on vizdoom.

---

> > ### Comment · Reviewer_BfC2 · 2021-11-22
> > **Concerns remain**
> >
> > I still have large concerns (see below). At least points 1 and 2 must be addressed to make this a useful paper, but I suspect that this will be impossible because the methods are equivalent (point 1) and the experimental results will not hold up (point 2). I would pivot away from D&E specifically and look more at reconciling options/skill learning work with intrinsic motivation and doing more evaluation and comparison of baselines in these areas.
> >
> > 1. Overlap with prior work
> > * "Compared with SMM, D&E does not need a target state distribution as input." SMM equation 3 terms (b) and (d) do not rely on a target state distribution, so D&E is essentially a subset of this method.
> > * I see that DIAYN is mentioned in the response to reviewer cJEK, but it doesn't clearly describe when DIAYN would fail. I don't think there is a real difference between D&E and DIAYN here, but if there is, some experiments or math would demonstrate this.
> >
> > 2. Results are not apples-to-apples comparisons
> > * "It is unreasonable to directly use all timesteps of D&E when plotting the best agent’s performance, for its training process does not directly use data from other agents." The data from other agents is used indirectly, and that means all timesteps should be counted.
> > * There at least needs to be a baseline that uses the same number of agents training in parallel (like IMPALA can).
> >
> > 3. Switching to ICM on VizDoom
> > * To make count work on VizDoom, pseudo counts can be used (e.g. feed images through a random network that outputs a smaller latent vector).
> > * At least also show D&E + ICM on MiniGrid. This would more clearly demonstrate the compatibility of D&E with multiple intrinsic rewards.

---

### Official Review · Reviewer_cJEK · 2021-11-03

**Correctness:** 3
**Technical Novelty And Significance:** 1
**Empirical Novelty And Significance:** 3
**Recommendation:** 5
**Confidence:** 4

**Main Review:**

The main strengths of this work is in its ability to naturally harness distributed computational infrastructure to solve computationally intractable exploration problems.  To the extent that this work addresses the underlying problem it would be impactful and significant.

The biggest weakness of this work is that it is difficult to tell if the results are from the particular intrinsic motivation, or if they are a side effect of using parallelism to run independent training runs. The experiments show best of the population of the agents in the proposed method, introducing a selection bias.  Thus we could expect it to outperform other methods even if the intrinsic motivation was ineffective.  To correct for this  the average agent of your method could be report, or the baselines could be allowed to take the max of take the max of n independent runs.

On a similar issue, the correction for the number of samples in Table 1 is good, but it could be that max of 3 short runs is generically better than 1 long run in this setting, if initialization matters significantly.   A more apples-to-apples comparison would run the duplicates like mentioned above.

The second biggest concern is that, even if the intrinsic motivation is necessary, the proposed mechanism of effect is increasing the diversity of the exploration.  If this is the case, then the method should be compared to a naive application of a diversity bonus like such as:
"Diversity is All You Need: Learning Skills without a Reward Function" Eysenbach, B. et al.
"The Emergence of Individuality in Multi-Agent Reinforcement Learning"  Jiang J. et al.

My third concern is that the system could generally get stuck. Concretely, the claim in the last sentence of section 4 saying that an agent would avoid other agents' failures cuts both ways, in that it would also avoid other agent's successes.  This could end up with two agents which can each only solve half of environments and do not want to explore to solve the rest of them because the other agent is more successful in those cases.


Minor:
* The decay rate in 4 looks like it would sometimes increase over time if the agent did not receive any intrinsic reward.  This may cause oscillation, and is a bit counterintuitive.  If this is intentional it should be justified and flagged for readers.
* in equation 4 why is the constant 1/U rather than U?
* In equation 5 and 6 you should define I
* At the start of the last paragraph on page 8 the sentence structure around the phrases "on one hand, we use... on the other hand, we replace..." is unclear, as I assumed from the sentence structure that the se ideas would be conflicting in some way.
* It seems like the studies mentioned in section 5.3 "Ablation Study" are not actually ablations, in that they do not remove parts of the methods, but they visualize the method and check its sensitivity to changes in it's parameters.  These are good studies to have, but I would not consider them ablations.
* In Figure 5 the key is not in numerical order, which makes the figure harder to read.
* in the last paragraph of section 4 the phrase "is considered convergence" seems to be a typo.

**Summary Of The Paper:**

This paper proposes a distributed algorithm for exploration in reinforcement learning, based off of the principle of "divide and conquer".  This is implemented as a multi-agent system where each agent is run on it's own node in parallel and given a custom intrinsic reward to motivate exploration and diversity.  These intrinsic rewards are built out of standard intrinsic reward methods for exploration, but are modified so each agent not only gets reward from it's own intrinsic motivation function but also gets reward by how their actions score with respect to the other agent's intrinsic motivation functions.  This motivates each agent to specialize to find the strategies the other agents' intrinsic motivation functions would see as novel.

**Summary Of The Review:**

I will weakly recommend rejection, as it is unclear the extent to which the effect of this method is from the parallelism v.s. the intrinsic  motivation, and it is unclear how this approach would compare to the standard diversity promoting intrinsic motivation methods.

---

> ### Author Response · Authors · 2021-11-21
> **Response to Reviewer cJEK**
>
> We sincerely thank reviewer cJEK for the helpful feedback and suggestions on our paper. We address the reviewer's concerns below.
>
> Q: "It is difficult to tell if the results are from the particular intrinsic motivation, or if they are a side effect of using parallelism to run independent training runs."
>
> A: We conduct several independent experiments with each baseline algorithm, and plot the result curve with 95% confidential interval in Fig. 2. To avoid confusion and show this clearly, we have updated our paper and plotted the best performance for baseline algorithms in Appendix B.2, which may illustrate the benefits are mainly from the particular intrinsic reward.
>
> Q: "The method should be compared to a naive application of a diversity bonus."
>
> A: The main novelties of D&E are: 1. We designed a method to transform a single-agent environment into a multi-agent environment, and each agent has its own intrinsic reward model. 2. Taking advantage of parallel computing to accelerate the exploration process. For DIAYN, it classifies the states by using the latent variable $z$ and uses mutual information as the loss, which hinders the agent from exploring the state that has not been fully explored and $I(X,Z)$ is little. While in D&E, the intrinsic reward used will keep promoting exploration while encouraging diversity. For "The Emergence of Individuality in Multi-Agent Reinforcement Learning", it is to solve exploration problems in the multi-agent environment, while we focus on the single-agent environment by flexibly expanding the number of agents to improve the efficiency of the algorithm, and the number of agents is variable according to our needs.
>
> Q: "The system could generally get stuck."
>
> A: Because we encourage agents to explore different paths from other agents, the system may get stuck in theory. However, this requires several conditions, singleton environment, greedy action selection, and so on. We hardly find this in practice, and we also take several tricks to avoid blocking. 1. The extrinsic reward should be much greater than the intrinsic reward, for example, 10 times. 2. The sampling of actions in PPO retains some randomness. 3. Adjusting the ratio of $\omega_o$ and $\omega_s$ can alleviate the blocking phenomenon.
>
> To other advice, we will make corrections in the new version of the paper.

---

> > ### Comment · Reviewer_cJEK · 2021-11-26
> > **Response to Review Response (1/2)**
> >
> > Thank you for your detailed response.  This addresses some of my concerns, but major concerns still persist.
> >
> > > Q: "It is difficult to tell if the results are from the particular intrinsic motivation, or if they are a side effect of using parallelism to run independent training runs."
> >
> > >A: We conduct several independent experiments with each baseline algorithm, and plot the result curve with 95% confidential interval in Fig. 2. To avoid confusion and show this clearly, we have updated our paper and plotted the best performance for baseline algorithms in Appendix B.2, which may illustrate the benefits are mainly from the particular intrinsic reward.
> >
> > Thank you for including this.  This removes a major confounder from the experiments, and makes me confident that the states results are a result of the intrinsic motivation intervention.  Unfortunately, I do not feel I can raise my score because of concerns other reviews raised (listed below).
> >
> > > Q: "The method should be compared to a naive application of a diversity bonus."
> >
> > >... For DIAYN, it classifies the states by using the latent variable  and uses mutual information as the loss, which hinders the agent from exploring the state that has not been fully explored and  is little. While in D&E, the intrinsic reward used will keep promoting exploration while encouraging diversity.
> >
> > I agree that this is a failure mode of DIAYN, I think that all known diversity approaches have such failure modes (including this approach, as mentioned in my other concern).  I do not think such failure modes disqualify a method unless we have a method without any such failures, or if we can show that the failure modes of DIAYN are empirically much more of an issue than those of D&E, which would require an experimental comparison.
> >
> > >For "The Emergence of Individuality in Multi-Agent Reinforcement Learning", it is to solve exploration problems in the multi-agent environment, while we focus on the single-agent environment by flexibly expanding the number of agents to improve the efficiency of the algorithm, and the number of agents is variable according to our needs.
> >
> > It's true that this paper is about multi-agent exploration, but you essentially translate a single-agent problem into a multi-agent problem, putting methods like this in-scope.  I think even if this multi-agent method outperforms your intrinsic reward, I think the idea of making the single agent setting into a multi-agent setting, and using a multi-agent exploration method is a novel approach on it's own.  This comparison would be to identify if the diversity is the cause of the gains, or if there is something specific about this intrinsic reward that makes it work, which is important to identify what the contribution is.
> >
> > >Q: "The system could generally get stuck."
> >
> > >A: Because we encourage agents to explore different paths from other agents, the system may get stuck in theory. However, this requires several conditions, singleton environment, greedy action selection, and so on. We hardly find this in practice, and we also take several tricks to avoid blocking. 1. The extrinsic reward should be much greater than the intrinsic reward, for example, 10 times. 2. The sampling of actions in PPO retains some randomness. 3. Adjusting the ratio of and can alleviate the blocking phenomenon.
> >
> > I do not understand the "singleton environment" condition.  But It does seem like this could be a problem that shows up reasonably often in practice, and I am not convinced it is not happening in the current experiments.  I think the method may be useful even with this vulnerability, but I would need to see it compared to other methods so I could judge how this issue compares to the problems with other diversity methods.  Without these experiments it is difficult for me to raise my score further.

---

> > ### Comment · Reviewer_cJEK · 2021-11-26
> > **Response to Review Response (2/2)**
> >
> > In reading the other reviews I also find a few points convincing which stops me from upgrading my score more:
> > From Reviewer BfC2:
> > >* The results are not apples-to-apples comparisons. Figures 2 and 5 and Tables 1 and 2 should show the total number of steps across all agents for each method. Since D&E uses 3 agents, its performance is effectively reported as being 3 times more sample efficient than it actually is.
> > > * Using 3 agents in parallel is not very scaled. For example, IMPALA is frequently run with 40+ actors. If the motivation for D&E is parallelism, then I’d expect more agents to be used.
> >
> > I think these are both very good points.  The first point is particularly worrying, and is why I did not raise my score.  I see where you are coming from on comparing in this way, since it measures the direct experience the networks get to train, but at the end of the day environment samples are the large bottleneck, and using 3x the samples is a problem.  I think that realization makes your approach start off behind, however it could still be the case that it will make it up in the long run by reaching a higher ceiling.  However, to tell if this is the case the baselines would have to be run 3x as long.
> >
> > The other points I found convincing were
> > From Reviewer VtVT:
> > >Seemingly missing references/discussion regarding some extremely relevant related work. Most closely related is [1] as it addresses the same setting of multiple concurrent agents solving a single-agent task by a divide-and-conquer exploration approach. Moreover, [2] introduces the idea of multiple agents each maintaining their own "novelty detection" function and combining these functions to generate intrinsic rewards that prevent redundant exploration of similar spaces.
> >
> > From Reviewer xEFb:
> > >The authors mention that they used the default hyperparameters for the baseline methods, but as some of the methods were not originally benchmarked on VizDoom (e.g. AMIGo and RAPID), it seems that these VizDoom baselines may be improperly tuned.

---

### Decision · Program_Chairs · 2022-01-20

**Decision:**

Reject

**Comment:**

The paper proposes a strategy for multiple learning agents to explore a large RL problem's state space, via the divide and conqeuer principle. It prescribes a design for each agent's reward function, which when optimized enables the agents to 'carve out' and cover different parts of the state space yielding efficient exploratory behavior. The argument for efficacy of the proposed method is purely experimental, with numerical benchmarking on complex simulated environments.

The reviewers have raised several concerns that persist even after receiving detailed responses from the author(s). These include the lack of discussion about comparisons with seemingly closely related and applicable work, the perception that the comparisons of this method with others are not fair ("not apples to apples"), and the assessment that the ablation studies and investigation of the sensitivity to hyperparameters may not be comprehensive to make a compelling argument. Thus, keeping in mind the unanimous impression of the reviewers, I am of the view that while the paper contributes an interesting principle, more work is needed to argue for its acceptance in a clear way.